# Bipedal Static Supination and Dynamic Forefoot Loading Characteristics in Taiwanese College Badminton Players: A Cross-Sectional Study

**DOI:** 10.3390/bioengineering10040498

**Published:** 2023-04-21

**Authors:** Tong-Hsien Chow, Chin-Chia Hsu, Chih-Cheng Chen, Chin-Hsien Hsu

**Affiliations:** 1Department of Sports Science, R.O.C. Military Academy, Kaohsiung 830208, Taiwan; thchowma@gmail.com; 2Department of International Business, Ming Chuan University, Taipei 11103, Taiwan; 3Department of Sport Management, Aletheia University, New Taipei City 25135, Taiwan; au1692@au.edu.tw; 4Department of Leisure Industry Management, National Chin-Yi University of Technology, Taichung 41170, Taiwan

**Keywords:** badminton, arch index (AI), plantar pressure distributions (PPDs), centers of gravity, rearfoot postural alignment, supinated foot

## Abstract

Context: Badminton is a unilateral sport that involves repetitive jumping, lunging and quick changes of direction with the lower limb, thus, plantar pressure profiles and foot postural profiles are critical to maintaining balance and coordination. Objective: The purpose of this study was to explore the characteristics of static and dynamic plantar pressure profiles with rearfoot posture in elite and recreational badminton players as well as assess the transitional changes of plantar loads between static and dynamic states. Methods: A cross-sectional survey was conducted among 65 college-level elite male badminton players (mean age: 20.2 ± 1.2 years; mean height: 177.4 ± 4.6 cm; mean weight: 72.6 ± 4.6 kg) and 68 recreational badminton players of the same gender (mean age: 19.9 ± 0.8 years; mean height: 170.3 ± 3.9 cm; mean weight: 67.7 ± 3.2 kg). The JC Mat was used to evaluate the arch index (AI), plantar pressure distribution (PPD), centers of gravity, and the characteristics of the footprint. Static foot posture was determined by examining the rearfoot alignment. Results: Both groups’ AI fell within the normal range. The static plantar loads of the elite group were distributed at the bipedal lateral part of longitudinal arches and heels (*p* < 0.01), while the right foot experienced higher centers of gravity (*p* < 0.05). The elite group’s static rearfoot postural alignment exhibited a higher degree of rearfoot varus than the recreational group (*p* < 0.05). In addition, the elite group’s dynamic plantar loads were mainly exerted at the medial and lateral metatarsals of both feet (*p* < 0.05). During the transition state, the recreational group’s plantar loads were mainly shifted to the bipedal lateral part of metatarsals and heels (*p* < 0.05), whereas the elite group’s bipedal lateral longitudinal arches as well as the medial and lateral heels experienced a reduction in plantar loads (*p* < 0.01). Conclusion: For elite badminton players, the findings revealed a possible connection among the static supinated foot, centers of gravity tending towards the right foot, and increased forefoot plantar loads in the dynamic state. The finding merits further exploration of the possible links between transitional changes in plantar pressure distribution in both states and related foot injuries resulting from intense competition and regular training in badminton.

## 1. Introduction

Badminton players often have frequent jumps, lunges, and quick changes of direction during competitions or regular training, which makes their feet and lower limbs need to repeatedly bear the vertical ground reaction force equivalent to 2.1 to 2.5 times their body weight [1,2,3]. Therefore, badminton injuries are more common than in other sports, accounting for about 1–5% of all sports injuries [4,5]. In particular, it is prone to overuse and fatigue pain in the feet and lower extremities [3]. As maintaining balance in badminton is one of the important factors to prevent injury [6], thus, lower extremity asymmetry should be anticipated and considered when planning injury prevention strategies in badminton [7].

With regards to the symptoms caused by performing repeated foot movements in these exercises, the cuboid syndrome most commonly affects runners, tennis and basketball players, and ballet dancers [8,9]. The common feature of these sports is that the changes of direction are rapid and the athlete needs to exert a great deal of force through their feet [9]. As a result, the cuboid joint and its surrounding attachments are thus placed under additional strain. Repeated pressure exerted on the cuboid joint may lead to a loosening of the support, causing cuboid joint dislocation or displacement [9].

The arch structure of the foot acts as a shock absorber for the human body to buffer the pressure on the plantar of the foot during running or jumping [10]. The medial longitudinal arch (MLA) is one of the foot arches which provides sufficient elastic and twisting forces to absorb the ground reaction forces, thereby weakening impact, preventing injury, and delaying fatigue [11]. The changes in MLA structure may alter the biomechanics of the lower extremity, resulting in altered plantar pressure in people with podiatrics or other articular pathologies of the lower extremities [12,13,14]. Thus, atypical foot shape or repetitive plantar loading are often associated with lower extremity and foot injuries, particularly those caused by running [15]. In addition, recent studies have found that changes in the structure of the MLA and the rearfoot valgus angle serve as reliable predictors of midfoot and rearfoot pressure-time integrals as well as the rearfoot posture in healthy runners [16]. The fact is well known that the arch index (AI) from footprints has been widely accepted as a reliable and relatively accurate method for determining MLA and arch height [17,18,19,20]. A static posture of the MLA is also a reliable method for evaluating dynamic foot function to determine whether a patient has foot-specific pathologies [21,22,23] or lower extremity dysfunction that might increase the risk of injury [24,25,26].

Plantar pressure measurements can be used to assess plantar loading, podiatric features, gait behavior, and rehabilitation conditions in subjects during walking and running [27,28,29]. Characterizing plantar pressure distribution may contribute to preventing lower extremity injuries [30] and understanding how the feet and ankles experience weight bearing and weight transfer during gait [31]. Parameters gained from the footprint can provide insight into the spatial relationships among multi-segments of the foot and its fine structure and functions [32], and these can be used in the detection, prevention, treatment, and rehabilitation of podiatric foot deformity recurrence [33,34]. As a result, measuring the plantar pressure profile is helpful in studying the biomechanics of the foot [35].

Several previous studies have explored foot pressure profiles and lower limb pain profiles in specific elite athletes based on static and dynamic plantar pressure measurements [36,37,38,39]. On this basis, in view of the fact that badminton is characterized by rapid displacement and instantaneous changes of direction, the features of static and dynamic plantar pressure profiles as well as the transitional changes between static and dynamic states are worth further exploration. Based on these arguments, the study aimed to explore the correlations among the arch index, plantar pressure distribution, centers of gravity balance, and rearfoot postural alignment of both feet in college-level elite badminton players during static stance and walking. It was hypothesized that badminton players exhibit unique foot pressure distribution and foot posture that may be related to the rapid displacement of footwork and balance maintenance of centers of gravity as compared to recreational badminton players.

## 2. Materials and Methods

### 2.1. Participants

This study is a cross-sectional survey of college-level elite badminton players conducted during their non-competition period. Participants in the study were 133 male college and university students recruited during their studies, and they were further divided into two groups: 65 elite badminton players (referred to as the elite group) and 68 recreational badminton players (referred to as the recreational group). Those eligible participants for inclusion in the elite group should be qualified first-class badminton athletes who had experienced competitions and training for more than five consecutive years at the Taiwan University Badminton Super Cup Competition, College Individual Championship Badminton Competition, and Badminton Division Championship of the National Intercollegiate Athletic Games in Taiwan. The recruitment of the elite badminton players experienced an approximately 20% dropout rate (65/81) attributed to (1) their attendance rates and (2) certificates of previous fractures and surgeries from hospitals. Exercise schedules for these elite badminton players included basic movements and aerobic exercises between 10 A.M. and 12 A.M. Training times for weights and tactical exercises were scheduled between 2 P.M. to 4 P.M. Regular sprinting for 1 to 2 h was scheduled every 2 to 4 days a week. 

As controls in this study, 68 eligible recreational badminton players were selected from 88 participants who had played badminton at least twice a week at badminton courts or sports fields within the past 6 months and also had at least 3 years of recreational badminton experience. Approximately 23% of recreational participants were dropped from the recruitment process mainly based on the following factors: (1) their attendance rates; (2) having professionally trained or competed in another sport; (3) certificates of previous fractures and surgeries from hospitals. For both groups, each participant in the present study was confirmed to hold the racket with the dominant right hand. Participants were excluded if they had previously undergone lower-limb surgery, lower-limb dislocations, or fractures in the preceding six months. In addition, other lower extremity musculoskeletal diseases, including leg length discrepancies, foot calcaneal spurs, skeletal rheumatoid arthritis, and lower extremity neuropathies were also listed as elimination conditions. These exclusion conditions mentioned above are mainly confirmed to be derived from the medical certificate report provided by each participant, or the individual-related sports or training injury records provided by coaches and athletic trainers at their school. Furthermore, considering that several studies have demonstrated in the past that body weight affects arch and plantar pressure characteristics, particularly the association between obesity and flat feet in adults and children [20,40,41,42], body weight may be a factor affecting the arch and shape characteristics of the foot. Therefore, the study recorded basic physical characteristics (e.g., age, height, weight, body mass index (BMI)) as well as the training experiences of all participants. To qualify for participation, participants had to have a BMI between 18.5 and 24, and those outside this range were also excluded. An overview of the basic demographic characteristics of the participants is shown in Table 1, which demonstrates that the height, mass, BMI, and badminton training experiences of both groups are statistically significant differences after being tested by a two-group student-t test at 95% confidence level. During the course of this study, all research protocols were carried out in accordance with the ethical standards of the research ethics committee and with the Declaration of Helsinki.

### 2.2. Instruments and Equipment

The JC Mat optical plantar pressure analyser coupled with the FPDS-Pro program (JC Mat, View Grand International Co Ltd., New Taipei City, Taiwan) was applied to measure the parameters of bipedal arch index (AI) and plantar pressure distribution (PPD) as well as the centers of gravity balance of research participants [43]. The device has been proven to be repeatable and reproducible in experiments from previous studies on expert athletes’ AI value and PPD performance during static standing and dynamic walking [36,37]. The relevant characteristics and conditions of use of this equipment are based on the following factors: (1) each side of the sensing area (32 cm × 17 cm) is designed with 13,600 sensors for measuring fine plantar pressure; (2) sensitive pressure sensors with large sensing areas can mark and present a delicate plantar pressure profile with dense round dots; (3) both the spatial and temporal colour plantar footprints and real barefoot images can be recorded simultaneously; (4) arch index, toe angle, the pressure distribution of footprints, and centers of gravity balance can be measured immediately; (5) plantar pressure distribution and footprint images correspond with isostatic weight calibration throughout the sensing platform.

### 2.3. Plantar Pressure Assessment

To coincide with the athletes’ training courses, each experiment time was scheduled before their specialty training between 7 A.M. and 9 A.M. on the same day. Before the experiment, each participant was surveyed to record their demographic characteristics. Next, participants were asked to obey the following steps in order to obtain data on static footprints by completing brief upright standing trials: (1) Take off their shoes and socks first and roll their trousers above the knees. (2) Stand barefoot on the sensing range and measurement area above the JC Mat and maintain a natural posture with the feet shoulder-width apart. (3) Maintain a calm and relaxed body posture with arms naturally hung down and looking directly into the front experimenter’s eyes. (4) Hold a relaxed position while maintaining a balanced posture until the JC Mat detects no significant change in plantar pressure measurements. Once the participant completes step 4 of the protocol, the JC Mat automatically records the preliminary results of plantar pressure distribution from their static footprint. 

During the dynamic plantar pressure measurement procedure, participants were first asked to practice walking barefoot along a four-meter-long walkway with a built-in JC Mat at their own steady and comfortable pace [44,45,46], walking to the end with a natural gait and then turning back to the starting point. Upon entering the experiment, each participant will perform three rounds of back and forth walk trials on the walkway at their own steady and comfortable pace until the dynamic plantar pressure of each foot is correctly captured at least three times, i.e., each foot is fully stepped on the sensing pad of the JC Mat marked with the sensor range and measurement area. Meanwhile, the built-in FPDS-Pro program in the JC Mat will collect preliminary data regarding dynamic plantar pressure and the gait travel lines of each foot. The PPDs from the total contact of footprints during the midstance phase of walking were then analyzed and identified by researchers. Three analyses of each participant’s feet were performed in the process, and the average of the three results was used to determine the dynamic plantar pressure distribution.

### 2.4. Plantar Pressure Data Analysis

The analysis software (FPDS-Pro-V2 software, View Grand International Co, Ltd., New Taipei City, Taiwan) built into JC Mat can be used to calculate pressure detection data and display colour plantar footprints and real barefoot images of the static and dynamic states. Once all measurement experiments had been completed, the researcher used the analysis software to create the first vertical line between the base of the second toe and the central base of the heel in the footprint image. Meanwhile, the program then automatically generates four horizontal tangents between the front and rear of the footprint excluding the toes. Next, three equal-part regions (i.e., regions A, B, and C) and six equal-part subregions (i.e., subregions 1, 2, 3, 4, 5, and 6) were thus automatically derived from the tangents in the footprint image [36]. Footprint regions of A, B, and C are referred to as the forefoot, midfoot, and rearfoot regions respectively. From the three regions, the six subregions were further divided sequentially from anterolateral to posteromedial as follows: (1) the lateral metatarsal bone (i.e., LM), (2) the lateral longitudinal arch (i.e., LLA), (3) the lateral heel (i.e., LH), (4) the medial metatarsal bone (i.e., MM), (5) the medial longitudinal arch (i.e., MLA), and (6) the medial heel (i.e., MH) [36,37,38,39]. An illustration of these regions is shown in Figure 1. The plantar pressure units from the six subregions were calculated as a percentage of the relative load. Based on the three footprint regions, the calculation method of the AI in this study is inherited from previous studies, all of which are based on the AI ratio formula suggested by Cavanagh and Rodgers, that is, the ratio of the area of the middle third of the footprint to the area of the complete footprint except the toes, as follows: AI = B/(A + B + C) [19,36,37,38,39]. According to Cavanagh and Rodgers’ defined principles, a normal arch is defined as an AI value between 0.21 and 0.26. A high arch is defined as an AI below 0.21. A flat arch is defined as an AI value greater than 0.26.

### 2.5. Rearfoot Postural Assessment

Assessment of rearfoot postural alignment for each participant was conducted following plantar pressure measurement. Participants in this procedure were first instructed to stand stably on a platform that was 30 cm high, then maintain a natural posture and keep their feet about 12–15 cm apart. Meanwhile, a digital camera was used to capture an image of the rearfoot postural alignment of each participant (with a minimum screen resolution of 96-ppi and 754 pixels). Based on Ribeiro et al.’s literature [22], the method involving determining the static angle of the rearfoot is as follows: ensure that each participant’s feet are standing on a platform with a horizontal line and locate three anatomical points from bottom to top along the posterior surface of the legs: (1) the first point is marked at the center of the calcaneal tuberosity; (2) the second point is marked above the calcaneal center; and (3) the third point is marked at the center of the lower third of the calf. The Biomech 2019 posture analysis program (Loran Engineering SrL, EmiliaRomagna, Italy) in the computer will automatically generate two intersection lines from the three-point connections. The first solid line was created from the calcaneal center to the center of the lower third of the calf, thus representing the first standard straight line of the lower extremity. The second dotted line, referred to as the flip angle line, was drawn from the center of the calcaneal tuberosity to the calcaneal center. Through the software, the frontal alignment of a digital image was calculated to determine the static rearfoot alignment. As a result of the intersection of the straight lines and the dotted line, the angle between 0° and 5° is defined as a normal foot, <0° is a varus foot, and >5° is a valgus foot [47].

### 2.6. Statistical Analysis

The parameters of demographic characteristics and training experiences of all participants were described by descriptive statistics. In this study, all numerical data were presented as mean ± standard deviation (SD). The AI values and the PPD values for the three regions of the forefoot, midfoot, and rearfoot as well as the PPD values for the six subregions of both groups were compared by independent samples *t*-test. All statistical significances for this study were defined at *p* < 0.05 (marked with *) and at *p* < 0.01 (marked with **). Statistical analysis was performed using the SPSS software package (IBM SPSS Statistics 21.0, Somers, New York, NY, USA) for this study.

## 3. Results and Discussion

### 3.1. Arch Index

The results of arch index analysis found no significant differences between the two groups. (Table 2). Therefore, the study showed that the arch indexes of both groups were within the normal range, and each had its symmetry.

### 3.2. Three Regional Plantar Pressure Distributions under Static and Dynamic States

A percentage of the relative load was used to represent plantar pressure distributions. In this study, the elite group experienced greater midfoot loads and lower rearfoot loads during the static stance compared to the recreational group (*p* < 0.01). While during the walking midstance phase, the results showed that the relative loads of the elite group were mainly shifted to the bipedal forefoot region and relatively reduced on the rearfoot region (*p* < 0.01). Comparing the changes between the static stance and the walking midstance phase in each group, the relative loads of the recreational group were mainly distributed to the right forefoot region (*p* < 0.01). The relative loads of the elite group were significantly shifted to the bipedal forefoot region (*p* < 0.05), while decreased on the midfoot and rearfoot regions (*p* < 0.01) (Table 3).

### 3.3. Six Subregional Plantar Pressure Distributions under Static and Dynamic States

The detailed six subregional relative loads were derived from the three plantar regions. When the participants of the elite group were standing statically, the relative loads on the plantar were distributed symmetrically at the bipedal lateral part of longitudinal arches and heels (*p* < 0.01), while the loads were found to be lower at the bipedal medial heels (*p* < 0.01). Such results seem to echo the previous literature that a supinated foot posture in which plantar pressure is mostly concentrated on the lateral part of the foot may contribute to reducing contact time with the ground during running [48]. The study of Hasegawa et al. also mentioned that the greater the degree of heel inversion of the runner, the shorter the time the foot is in contact with the ground [48]. Runners with conditions of shorter ground contact times and higher frequency of foot inversion are considered to contribute to increasing the efficiency of running [48]. Furthermore, the supinated foot posture has also been associated with the anatomy of cuboid syndrome. Both conditions of the foot are also frequently seen in certain athletes, including runners, basketball players, and tennis players [8]. Sports like these involve large forces through the foot or quick movements in which players change directions rapidly. This, in turn, may cause more stress on the joints and capsules around the cuboid bone as a result of overtightening the surrounding muscles and ligaments. As a result of repetitive forces exerted on the cuboid bone, the attached joints or ligaments may be damaged and torn, allowing the cuboid bone to dislocate or dislodge [9].

As for the results of the walking midstance phase, the elite group’s relative loads were significantly shifted from the lateral foot to the entire forefoot region, such as the bipedal medial and lateral metatarsal bones (*p* < 0.05). The results, to a certain extent, reflected the fact that the human body is typically accustomed to striking with the forefoot when running barefoot [49]. However, in case studies of athletes’ gait behavior, Guettler et al. mentioned that in athletes performing various basketball footwork, such as simulated lay-up tasks, instant one-foot landing, side-to-side shuffle dribbles, and maximum effort sprint, the fifth metatarsal experienced a higher plantar load [50]. In similar movements such as lay-up landing and shuttle run, Yu et al. asserted that specialized basketball footwork resulted in greater peak forces and higher plantar loads exerted on the fifth metatarsal bone [51]. Williams et al. went further and documented that athletes typically have higher peak forces observed below the second and third metatarsals after an intensive endurance exercise [52]. However, it has been found that forefoot strikes in various sports disciplines can reduce vertical and knee loads in athletes during competition [49].

Based on the results of comparisons of the static stance and walking midstance phases in each group, the recreational group’s relative loads were changed and distributed at the bipedal lateral part of metatarsals and heels (*p* < 0.05). In addition, it was found that the relative loads at the bipedal lateral longitudinal arches, as well as the medial and lateral heels, were significantly lower in the elite group (*p* < 0.01) (Table 4). In terms of the recreational group, however, the results seem to confirm previous research showing that when badminton players lunge forward, plantar pressure is mainly exerted on the heel and lateral foot [3,53]. Likewise, the study noted that athletes’ metatarsal heads and lateral heels as well as the lateral part of the foot were often the regions that make the most contact with the ground in different footwork [53]. A similar study was conducted on basketball discipline by Chua et al., who noted that a high plantar load is often applied not only to the heel during take-off steps, but also to the forefoot and rearfoot areas upon landing as well [54]. Furthermore, athletic behaviors involve frequent sprints and rapid changes of direction, such as in basketball and tennis players, where the foot tends toward a supinated position [8]. As for the changes in the plantar pressure distribution of the elite group from static to dynamic, similar findings are rare in the other literature, and were only found to be comparable to those observed in a study conducted by Bisiaux and Moretto, which noted that runners had significantly lower medial heel peak pressures and relative impulses 30 min after fatigue running [55].

### 3.4. Centers of Gravity Balance

Participants’ centers of gravity distributions were described as a percentage of gravity. As compared to the recreational group, the elite group’s centers of gravity were exerted more on the right foot (*p* < 0.05) and lower on the left foot (*p* < 0.05) when standing in the static position (Table 5). The findings echo those from the study by Petrinović et al. which found that badminton players had statistically significant differences in the morphology of their upper leg circumferences and forearms on both sides of their bodies [56]. Nadzalan et al. also mentioned that there is an asymmetric kinematic balance between the dominant and non-dominant limbs in the badminton lunge [57]. On the other hand, Hu et al. argue that the forward lunge task results in reduced plantar loads distributed to the great toe of the athlete’s dominant leg compared to the left and right maximal forward lunge task [3]. These findings can therefore be attributed to the fact that badminton is a unilateral sport, where the players’ dominant limbs often move more accurately and faster than the opposite limbs, they are thus prone to an imbalance between the left and right sides of their bodies [58]. Consequently, the dominant limb of badminton players results in a heavier distribution of centers of gravity on the corresponding side of their foot.

### 3.5. Rearfoot Postural Alignment

According to the results of rearfoot postural alignment, it was found that the bipedal rearfoot angles of both groups fell within the normal range. Nonetheless, the elite group had significantly lower values for static rearfoot postural alignment on both feet than the recreational group (*p* < 0.05) (Table 6). In research on similar exercise behaviours, Klem et al. found that increased rearfoot pronation during cutting maneuvers often results in injuries among basketball players [59]. Czerniecki argued that the special requirements of the athletes are related to the tension and tropism of the muscles: as we know, the triceps surae is responsible for the rearfoot inversion and supination of the foot and providing stiffness to the tarsal joints during lower limb movements [60].

### 3.6. Static Footprint Characteristics

The footprint images were determined by averaging the results of the six subregional PPDs in static and dynamic states within each homogenized representative subject. The static footprint characteristics of the elite group illustrated that the greater pressure profiles were mainly distributed at the bipedal lateral longitudinal arches (Figure 1).

The study not only investigated the characteristics of static and dynamic plantar pressure profiles with rearfoot postural profiles in Taiwanese college badminton players but also assessed the transitional changes between static and dynamic states. In addition, this study is expected to provide information on the changes in plantar pressure distribution that recreational badminton players may experience when they become elite athletes through repeated training or competition. The findings of the present study showed that the elite badminton players experienced a bipedal supinated foot and exerted more centers of gravity on the right foot during static standing, and increased forefoot loading in dynamic states. During the transition state, the relative loads of the recreational group shifted to the bipedal lateral part of the plantar, while the relative loads of the elite group decreased significantly along the bipedal lateral longitudinal arches and the entire heels of the foot. 

In this study, the limitations may be attributed to a limited analysis of plantar loading patterns in 65 elite males and 68 same-gender badminton players aged 19 to 22 years from Taiwanese colleges or universities. In addition, the physical condition of the recruited participants selected for the study only considered that they should meet the normal range of BMI recommended by the World Health Organization (WHO). Other body composition parameters, such as body fat mass, skeletal muscle mass, percent body fat, lower extremity (or appendicular) lean body mass, etc., were not evaluated and surveyed in this study. Based on these factors, however, it may cause some difficulties in interpreting the results of this study and inevitably limit the possibilities for generalization. Furthermore, the study only included participants who held the racket with their dominant right hand, and the issue of whether the participant had a dominant/non-dominant leg was not considered within the research. Therefore, further studies not only need to rely on a considerable sample size to examine whether badminton players have dominant legs, but also consider using an accelerometer to determine the impact of dominant legs on the distribution of plantar pressure and external intensity of lower limbs during training and competition [61]. The results of this study preliminarily examine the features of static and dynamic plantar pressure profiles as well as the transitional changes between static and dynamic states in Taiwanese college elite and recreational badminton players. The study differs from previous studies in that it incorporates the consideration of centers of gravity balance and the changes in rearfoot postural alignment. As a result of this study, the characteristics of plantar pressure profiles and the footprints of elite badminton players can be used in the development and design of badminton boots or related sports orthotic insoles, which are expected to cushion the uneven distribution of plantar loads, improve shoe comfortability, and reduce the risk of sports-related injuries suffered by badminton players. With respect to the imbalance of centers of gravity in elite badminton players, the lunge squat exercises have been considered to be effective rehabilitation training for improving static and dynamic joint balance [62].

## 4. Conclusions

The college-level elite badminton players in this study were classified as having normal arches. The static plantar pressure profiles showed that bipedal plantar loads were symmetrically distributed at the lateral part of longitudinal arches and heels, coupled with more exerted centers of gravity on the right foot and a greater degree of rearfoot varus on both feet. During the walking midstance phase, their plantar loads were mainly shifted to the medial and lateral metatarsals of the entire forefoot. From the transitional changes of the plantar loads in the recreational group, it was found that the plantar loads were mainly shifted to the lateral part of the metatarsals and heels of both feet. While in the elite group, the plantar loads were notably reduced at the bipedal lateral longitudinal arches as well as the medial and lateral heels. This study not only highlights the changes in plantar pressure distribution that amateur athletes may experience when they become elite athletes through repeated training or competition but also reflects the appearance of static and dynamic plantar pressure distributions in the daily life of elite badminton players. Additionally, the possible links between transitional changes in plantar pressure distribution in both states and related foot injuries deserve further investigation.

## Figures and Tables

**Figure 1 bioengineering-10-00498-f001:**
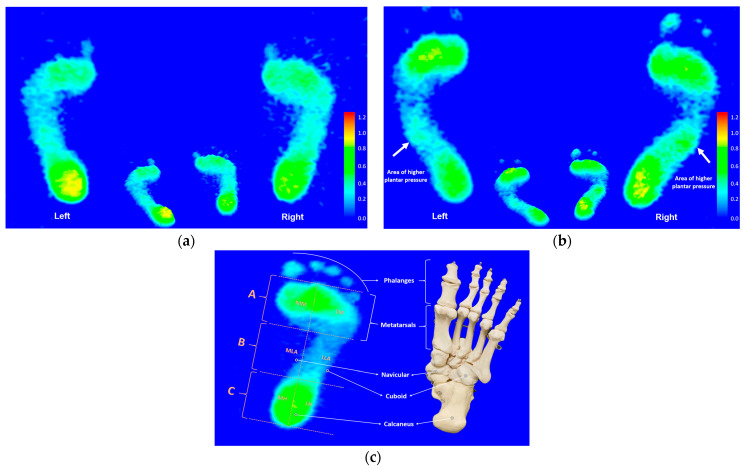
Static footprint image features of the representative participant in the (**a**) recreational group and the (**b**) elite group. Illustration (**c**) is the color footprint image and the corresponding foot anatomical structure model. White arrows indicate the areas of higher plantar pressure.

**Table 1 bioengineering-10-00498-t001:** Basic demographic characteristics of the participants.

Characteristic	Recreational Group (*n* = 68)	Elite Group (*n* = 65)
Age (years)	19.9 ± 0.8	20.2 ± 1.2
Height (cm)	170.3 ± 3.9	177.4 ± 4.6 *
Mass (kg)	67.7 ± 3.2	72.6 ± 4.6 **
BMI (kg/m^2^)	23.3 ± 0.4	23.1 ± 0.7 **
Badminton Training experience (years)	3.1 ± 0.7	5.9 ± 1.1 **

Abbreviation: BMI, body mass index (calculated as the weight in kilograms divided by the square of the height in meters). Note: Values are given as mean ± SD. * *p* < 0.05, ** *p* < 0.01 (student-*t* test, 2-tails).

**Table 2 bioengineering-10-00498-t002:** Bipedal arch indices of elite and recreational badminton players.

	Recreational Group (*n* = 68)	Elite Group (*n* = 65)	*p*-Value
Left foot	0.21 ± 0.06	0.20 ± 0.07	0.057
Right foot	0.21 ± 0.05	0.20 ± 0.06	0.116

The static bipedal arch indices of both groups are represented as mean ± SD. Statistical significance of *p*-values was determined by the independent sample *t*-test.

**Table 3 bioengineering-10-00498-t003:** Bipedal plantar pressure distributions of the three regions under static and dynamic states.

Region	Recreational Group (*n* = 68)	Elite Group (*n* = 65)
Static standing		
Left foot		
Forefoot (%)	21.37 ± 2.69	21.57 ± 2.34
Midfoot (%)	11.50 ± 10.62	12.27 ± 11.52 ^b^
Rearfoot (%)	17.13 ± 4.31	16.16 ± 8.10 ^b^
Right foot		
Forefoot (%)	21.69 ± 2.81	21.57 ± 2.90
Midfoot (%)	11.59 ± 10.16	12.17 ± 11.47 ^b^
Rearfoot (%)	16.73 ± 5.46	16.26 ± 8.21 ^b^
Midstance phase of walking		
Left foot		
Forefoot (%)	21.96 ± 2.44	26.80 ± 4.41 ^b,c^
Midfoot (%)	11.03 ± 9.96	9.74 ± 9.57 ^d^
Rearfoot (%)	17.01 ± 4.25	13.43 ± 7.43 ^b,d^
Right foot		
Forefoot (%)	22.19 ± 5.03 ^d^	25.30 ± 7.18 ^b,d^
Midfoot (%)	11.23 ± 10.62	10.63 ± 10.58 ^d^
Rearfoot (%)	16.86 ± 5.90	14.07 ± 7.70 ^b,d^

Bipedal plantar pressure distributions of the three regions under both states are represented as mean ± SD. Statistical significances of *p*-values were determined by the independent sample *t*-test. ^b^ *p* < 0.01, defined as statistically significant differences between both groups. ^c^ *p* < 0.05, ^d^ *p* < 0.01, defined as statistically significant differences between the static stance and the walking midstance phase in each group.

**Table 4 bioengineering-10-00498-t004:** Bipedal plantar pressure distributions of the six subregions under static and dynamic states.

Six Subregions	Static Standing		Midstance Phase of Walking	
	Left Foot	Right Foot	Left Foot	Right Foot
Recreational group (*n* = 68)				
Lateral Metatarsal bone (LM)	22.01 ± 2.80	22.05 ± 2.81	23.32 ± 1.65 ^d^	22.95 ± 5.18 ^c^
Lateral Longitudinal Arch (LLA)	21.67 ± 4.14	21.58 ± 1.86	20.58 ± 4.63	21.00 ± 5.74 ^d^
Lateral Heel (LH)	20.61 ± 1.80	20.69 ± 3.97	20.68 ± 4.56 ^d^	20.71 ± 5.51 ^c^
Medial Metatarsal bone (MM)	20.74 ± 2.44	21.33 ± 2.78	20.80 ± 1.77	21.42 ± 4.79
Medial Longitudinal Arch (MLA)	1.32 ± 0.41	1.60 ± 1.38	1.29 ± 0.40	1.45 ± 0.36
Medial Heel (MH)	13.66 ± 3.13	12.77 ± 3.55	13.23 ± 2.77	12.94 ± 3.00
Elite group (*n* = 65)				
Lateral Metatarsal bone (LM)	20.62 ± 2.42	21.31 ± 2.61	26.40 ± 2.26 ^a^	26.28 ± 7.17 ^a^
Lateral Longitudinal Arch (LLA)	23.58 ± 2.84 ^b^	23.42 ± 2.87 ^b^	18.63 ± 4.89 ^d^	20.28 ± 6.02 ^d^
Lateral Heel (LH)	23.85 ± 2.54 ^b^	24.01 ± 2.57 ^b^	19.87 ± 5.00 ^d^	20.36 ± 5.84 ^d^
Medial Metatarsal bone (MM)	22.51 ± 1.82	21.84 ± 3.17	27.19 ± 2.51 ^a^	24.33 ± 7.11 ^b^
Medial Longitudinal Arch (MLA)	0.97 ± 0.33	0.93 ± 0.32	0.85 ± 0.32	0.98 ± 0.34
Medial Heel (MH)	8.47 ± 2.36 ^b^	8.50 ± 2.65 ^b^	7.00 ± 1.44 ^b,d^	7.79 ± 2.29 ^d^

Bipedal plantar pressure distributions of the six subregions under both states are represented as a percentage of relative load and the values are expressed as mean ± SD. Statistical significances of *p*-values were determined by the independent sample *t*-test. ^a^ *p* < 0.05, ^b^ *p* < 0.01, defined as statistically significant differences between both groups. ^c^ *p* < 0.05, ^d^ *p* < 0.01, defined as statistically significant differences between the static stance and the walking midstance phase in each group.

**Table 5 bioengineering-10-00498-t005:** Centers of gravity balance assessment under static standing in elite badminton players.

	Recreational Group (*n* = 68)	Elite Group (*n* = 65)
Left foot	50.83 ± 5.25	47.08 ± 9.41 *
Right foot	49.17 ± 5.25	52.92 ± 9.41 *

The statistics of the percentage of bipedal centers of gravity are expressed as mean ± SD. Statistical significance of *p*-values was determined by the independent sample *t*-test. * *p* < 0.05, defined as statistically significant differences between both groups.

**Table 6 bioengineering-10-00498-t006:** Assessment of static rearfoot postural alignment in elite badminton players.

	Recreational Group (*n* = 68)	Elite Group (*n* = 65)
Left foot	4.04 ± 2.03	1.13 ± 0.64 *
Right foot	4.57 ± 3.15	1.02 ± 0.53 *

The statistics of static rearfoot postural alignment are represented as an angle (°) and the values are expressed as mean ± SD. Statistical significance of *p*-values was determined by the independent sample *t*-test. * *p* < 0.05, defined as statistically significant differences between both groups.

## Data Availability

The datasets generated and/or analyzed during the current study are available from the corresponding author upon reasonable request.

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
