# Peer review of "Bipedal Static Supination and Dynamic Forefoot Loading Characteristics in Taiwanese College Badminton Players: A Cross-Sectional Study"

_bioengineering, 2023, doi:10.3390/bioengineering10040498_

Round 1

Reviewer 1 Report

Title: please mention the study design

Abstract: Well written

Intro: Reads well. The purpose is well explained and the supporting literature has been mentioned clearly. The intro is however long and needs to be shortened substantially

Methodology: Clearly explained. There are grammatical and typographical errors throughout the manuscript which need to be modified. Again, needs to be substantially shorter

Results: Well represented. Tables are legible.

Discussion: Apart from some typographical and grammatical errors, overall well written. Can be presented under different subheadings

Conclusion: Clearly written

Author Response

Dear Reviewer,
We sincerely appreciated the reviewers’ constructive suggestions and constructive comments on our manuscript (ID: bioengineering-2283403). The suggestions and comments are helpful for improving our manuscript. We are submitting the revised version of the manuscript with our responses to the suggestions and comments by the reviewers. 

Our responses to each suggestion and comment are written in detail in the attached revised manuscript, and they are also presented in red text with a grey background color in the revised manuscript.

  1. Title: please mention the study design

Response:

Thank you for the suggestion. We refined our paper by adding the study design in the Title of the manuscript. The modifications are as follows:

Line 3~4, Bipedal Static Supination and Dynamic Forefoot Loading Characteristics in Taiwanese College Badminton Players: A Cross-Sectional Study

2. Intro: Reads well. The purpose is well explained and the supporting literature has been mentioned clearly. The intro is however long and needs to be shortened substantially

Response:

Thank you for the suggestion. We refined our paper by making concise revisions in the Introduction of the revised version. The modifications are as follows:

Line 43~51, Badminton players often have frequent jumps, lunges and quick changes of direction during competitions or regular training, which makes their feet and lower limbs need to repeatedly bear the vertical ground reaction force equivalent to 2.1 to 2.5 times their body weight [1-3]. Therefore, badminton injuries are more common than in other sports, accounting for about 1-5% of all sports injuries [4,5]. In particular, it is prone to overuse and fatigue pain in the feet and lower extremities [3]. Because maintaining balance in badminton is one of the important factors to prevent injury [6], thus, lower extremity asymmetry should be anticipated and considered when planning injury prevention strategies in badminton [7].

Line 59~67, The arch structure of the foot acts as a shock absorber for the human body to buffer the pressure on the plantar of the foot during running or jumping [10]. The medial longitudinal arch (MLA) is one of the foot arches which provides sufficient elastic and twisting forces to absorb the ground reaction forces, thereby weakening impact, preventing injury and delaying fatigue [11]. The changes in MLA structure may alter the biomechanics of the lower extremity, resulting in altered plantar pressure in people with podiatrics or other articular pathologies of the lower extremities [12-14]. Thus, atypical foot shape or repetitive plantar loading are often associated with lower extremity and foot injuries, particularly those caused by running [15].

Line 76~80, Plantar pressure measurements can be used to assess plantar loading, podiatric features, gait behavior and rehabilitation conditions in subjects during walking and running [27-29]. Characterizing plantar pressure distribution may contribute to preventing lower extremity injuries [30] and understanding how the feet and ankles experience weight bearing and weight transfer during gait [31].

Line 86~89, Since several previous studies have explored foot pressure profiles and lower limb pain profiles in specific elite athletes based on static and dynamic plantar pressure measurements [36-39]. On this basis, in view of the fact that badminton is characterized by rapid displacement and instantaneous change of direction.

Reviewer 2 Report

General Comments

This is an interesting study, evaluating the characteristics of static and dynamic plantar pressure profiles with rearfoot posture in young male elite (n=65) and recreational (n=68) badminton players with assessing the transitional changes of plantar loads between static and dynamic states. The JC Mat optical plantar pressure analyser was used to evaluate the arch index, plantar pressure distribution, centres of gravity and the characteristics of the footprint. Static foot posture was determined by examining the rearfoot alignment. The results of this study demonstrated that the elite group’s static rearfoot postural alignment exhibited a higher degree of rearfoot varus than the recreational group, whereas the elite group’s dynamic plantar loads were mainly exerted at the medial and lateral metatarsals of both feet. During the transition state, the recreational group’s plantar loads were mainly shifted to the bipedal lateral part of the metatarsals and heels, whereas the elite group’s bipedal lateral longitudinal arches as well as the medial and lateral heels experienced a reduction in plantar loads.

1.The authors should be to describe more detail the exclusion criteria of participants.  How was controlled that the participants did not have:

- Pathological lower extremity abnormalities - pes planus (flat feet), genu varum (bowleg), genu valgum (knock knees)? - Diabetes, hypertension, pain, incl. low back pain?  

2. In my opinion, it is obligatory to add as a limiting factor of the study the facts, that:  (a) only males were measured, and (b)  body composition (fat distribution across the body, lower extremity (or appendicular) lean mass, muscle mass of the lower extremities, etc.) was not assessed in this study which cause some difficulties to interpret the results of this study. This notice  should be mentioned and analysed at the end of the Discussion.

In conclusion, I recommend to accept this manuscript for publication after minor corrections.

Specific Comments

Abstract

Please add information that the subjects were males.

2. Materials and Methods

2.1 Participants

Page 3. Please describe more detailed the exclusion criteria for participants (see General Comments).

Page 4. Table 1. Please correct the unit of BMI in Table 1: (kg/m²) instead of (m/kg).

3. Results

Page 7. Table 4. Please indicate that bipedal plantar pressure distributions of the six subregions under both states are represented as a percentage of relative load.

Page 8. Table 6. Please indicate that the rearfoot postural alignment is represented as an angle (°)

4. Discussion

Page 10, Lines 400-410. Please describe more limitations of this study at the end of Discussion (see General Comments).

Author Response

Dear Reviewer,
We sincerely appreciated the reviewers’ constructive suggestions and constructive comments on our manuscript (ID: bioengineering-2283403). The suggestions and comments are helpful for improving our manuscript. We are submitting the revised version of the manuscript with our responses to the suggestions and comments by the reviewers. 

Our responses to each suggestion and comment are written in detail in the attached revised manuscript, and they are also presented in red text with a grey background color in the revised manuscript.

Reviewer 3 Report

Authors compared the foot structure and plantar loading in elite and recreational badminton players, while the background literature was not enough to justify the rationale and background of this study, for example, why the typically developed non badminton playing cohort was not selected to check if the difference is from badminton playing or any other potential factors.

The height, mass and BMI showed difference for the two groups, just wondering the difference in foot structure is because the above anthropometric factors or badminton playing and training experience?

Please be specific for the dynamic tasks being performed.

Please include figures to illustrate the analysis of foot structure.

Why select walking gait as example, not other badminton-specific tasks?

In the conclusion section, apart from the summary, if there is any practical implications?

Author Response

(The authors gave the same response as above.)

Round 2

Reviewer 3 Report

After reading the revision and response letter, authors still failed to address the previous concerns for this manuscript.